# Sparse Causal Model: A Novel Approach for Causal Discovery and Attributions on Sparse Dataset

## Abstract

This paper introduces a novel approach to tackle the challenges of causal modeling and attribution in sparse and non-continuous data with limited feature knowledge. Traditional methods rely on static inputs and lack adaptability to dynamic changes in causal relationships, resulting in a limited understanding and goodness-of-fit. To address this challenge, we introduce a unique causal discovery framework on real-world sparse datasets. We leverage a Directed Acyclic Graph (DAG) by discovering causal relationships between the variables by identifying confounder-treatment pairs that make the variable selection process robust and efficient. We propose a three-stage causal model that uses multiple distinct regressors such as likelihood-based, tree-based, and Generalized Additive Models (GAMs). Furthermore, we introduce a Model Score by including the sensitivity analysis involving random shuffling confounders and treatments to select the best optimal model. We implement a partial dependency approach to understand the attribution of variables, contributing by adding a 53% increase in $R^2$ score compared to traditional methods. This research underscores the limitations of conventional approaches in addressing real-world challenges to address practical scenarios effectively.

## 1 Introduction

Understanding causal relationships is crucial for informed decision-making, but traditional methods struggle with sparse datasets, leading to inaccuracies in identifying causes and constructing Directed Acyclic Graphs (DAGs). This highlights the need for more robust techniques to ensure reliable causal inferences in such challenging scenarios. This paper introduces a causal discovery framework, termed the "union" framework, which seeks to address these challenges by integrating existing domain knowledge with the discovery of new indicative causal pairs. This approach leverages a combination of both parametric and non-parametric models, enhanced by advanced sensitivity analyses. By incorporating randomized shuffling of confounders and treatment variables, our methodology ensures the identification of robust causal models, thereby enhancing the accuracy and reliability of the derived inferences Rohrer (2018); VanderWeele (2019). Furthermore, we propose a comprehensive methodology for attributing outcomes to both direct and indirect effects of causal variables, while separating the baseline effects. This is achieved through a detailed algorithm that applies scaling techniques, allowing for precise estimation of variable attributions.

One significant limitation of existing methodologies is their inability to accurately attribute direct and indirect effects. These approaches often struggle to effectively disentangle the influence of primary variables—those directly affecting the outcome—from secondary variables with indirect causal relationships Peters et al. (2016); VanderWeele (2019). To address this shortcoming, we implemented an output decomposition model that removes seasonality, trends, and other extraneous effects, thereby constructing a robust baseline model. Moreover, current methodologies for discovering causal relationships are particularly inefficient when dealing with sparse observations. To the best of our knowledge, there are no existing methods that effectively incorporate or backpropagate the effects of sensitivity treatments to identify the optimal combination of models for causal inference.

## 2 RELATED WORK

Causal inference begins with the specification of essential assumptions, such as domain-specific assumptions, followed by the construction of a causal graph. This process involves identifying confounder pairs, attributing causal influences, diagnosing the causal structure, and estimating effects Shao et al. (2017); Sharma et al. (2021); Ibeling & Icard (2024); Zheng et al. (2024). Traditional machine learning algorithms used for effect estimation include the average treatment effect (ATE) and the conditional average treatment effect (CATE), both facilitating the estimation of causal effects. Moreover, these methods address the issue of confounding effects prevalent in observational studies, thereby ensuring more accurate treatment effect estimation.

In prior research, Colnet et al. (2024) underscores a critical limitation in existing causal discovery methods, namely their inability to control the family-wise error rate, which increases the likelihood of erroneous identification of causal relationships. Furthermore, emphasizes the significant advantage of invariant causal predictors over non-causal predictors, noting that the influence of invariant causal predictors on the outcome variable remains stable despite environmental changes. Peters et al. (2016); Bergstra & Bengio (2012); Frauen et al. (2023); Zheng et al. (2024) also addresses limitations in the field, particularly highlighting the challenges associated with relying on randomized controlled trials (RCTs). Additionally, Peters et al. (2016) caution that large observational studies, while offering high representativeness, may inadvertently introduce biases by conflating confounding effects with the treatment of interest.

## 3 PROBLEM FORMULATION

The objective of a sparse causal model is to identify the causal relationships among variables and attribute both direct and indirect effects of these variables on the outcome. Given a sparse input dataset $D = (x_1, x_2, \ldots, x_m, Y)$, where $X_{\text{input}} = \{x_1, x_2, \ldots, x_m\}$ represents the set of input variables and $Y$ denotes the outcome variable, the task is to determine the corresponding adjacency matrix that captures these causal relationships. In addition, the goal is to construct a robust causal model capable of accurate output prediction, along with quantifying the respective attributions of each variable to the outcome.

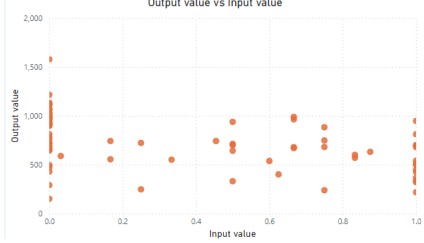

Figure 1: Distribution of sparse. input versus output.

## 4 METHODOLOGY

We propose a methodology that leverages transfer function-based estimation within a rigorously constructed causal model, incorporating causal discovery. This approach involves analyzing the causal directed acyclic graph (DAG), estimating sensitivity ($R^2$) scores, and evaluating the impact of baseline metrics alongside causal variables. Our study offers a solution to the challenges of causal modeling and attribution in sparse, feature-limited, and non-continuous datasets. Unlike traditional methods that rely on static inputs, our approach adapts to dynamic changes, providing a deeper understanding of causal relationships.

### 4.0.1 CONFIGURATION FRAMEWORK

We developed a causal discovery model integrating three machine learning algorithms: Linear Regression, XGBoost, and Generalized Additive Models (GAM). Hyperparameter tuning using Ran-

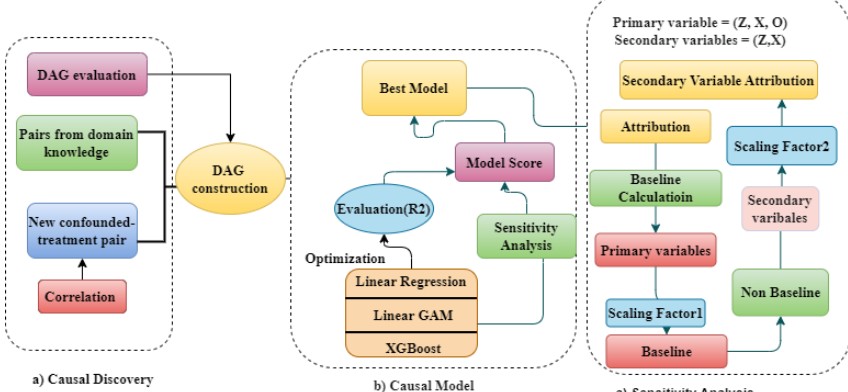

Figure 2: Overview of our proposed framework, Sparse causal model, consisting of three stages: a) Causal discovery; b) Causal model; and c) Sensitivity Analysis.

domizedSearchCV Bergstra & Bengio (2012) was performed, which is crucial for optimizing models in diverse applications Gambella et al. (2021); Bergstra et al. (2011). The best hyperparameters were configured using scikit-learn Pedregosa et al. (2011). For training samples $(x_1, y_1), \ldots, (x_i, y_i)$, the conditional expectation in linear regression is expressed and identified the optimal model by minimizing the loss across all training samples:

$$E(y|x) = \int y \frac{f(x, y)}{f(x)} \, dy$$

$$f^* = \arg\min_{f \in F} \frac{1}{n} \sum_{i=1}^{n} L(f(x_i), y_i)$$

XGBoost minimizes the following objective function Yang & Shami (2020); Chen & Guestrin (2016) where $t$ is the number of leaves, $G$ and $H$ are the first and second-order gradient sums, and $\gamma, \lambda$ are penalty coefficients. GAM approximates regression by summing smooth functions of predictor variables Hastie (2017); Saltelli et al. (2006); Shao et al. (2017)

$$\text{Obj} = -\frac{1}{2} \sum_{j=1}^{t} \frac{G_j^2}{H_j + \lambda} + \gamma t$$

$$y_i = f_1(x_{1,i}) + f_2(x_{2,i}) + \cdots + f_n(x_{n,i}) + \epsilon_i$$

Our methodology employs a data-driven approach, starting with correlation analysis to identify potential causal relationships, followed by constructing treatment models for each pair to account for confounders. Covariates were defined as all variables excluding the target outcome and treatment, and residuals were stored after predicting treatment values. A baseline outcome model was fitted using covariates and control variables, followed by a model excluding these controls. Residual models predicted outcome residuals based on treatment model residuals. These models were integrated into a causal framework to estimate treatment effects, with sensitivity analyses through random treatment shuffling and confounder introduction to ensure robustness.

### 4.0.2 IDENTIFICATION OF THE CONFOUNDER-TREATMENT PAIR

Selecting appropriate confounder-treatment pairs is crucial in causal inference Rohrer (2018). This process involves determining which covariates should be controlled for confounding, a task that typically depends on assumptions about causality and the variables of interest. Given the challenges in identifying pairs without prior domain knowledge, our study introduced a data-driven selection method for confounded treatment pairs by analyzing correlation path coefficients.

$$\text{corr}(X_k, X_l) = \frac{\text{cov}(X_k, X_l)}{\sigma_{X_k} \sigma_{X_l}}$$

$$= \frac{\sum_{i=1}^{n} (X_{k,i} - \overline{X_k})(X_{l,i} - \overline{X_l})}{\sqrt{\sum_{i=1}^{n}(X_{k,i} - \overline{X_k})^2} \cdot \sqrt{\sum_{i=1}^{n}(X_{l,i} - \overline{X_l})^2}} \tag{1}$$

We select the top $k$ pairs based on correlation coefficients and exclude perfectly correlated pairs ($\text{corr}_{kl} = 1$) to preserve the utility of the Directed Acyclic Graph (DAG). This method improves causal inference by systematically identifying significant confounder-treatment pairs. To maintain the integrity and utility of the Directed Acyclic Graph (DAG) for causal inference, any pairs exhibiting perfect correlations ($\text{corr}_{kl} = 1$) are excluded, as they do not contribute additional information.

## 4.1 CAUSAL DIRECTED ACYCLIC GRAPH CONSTRUCTION

Let the dataset be denoted as $D$ with $n$ observations, the outcome variable be $Y$, the treatment variable be $X$, and the confounder variables be $Z_1, Z_2, \ldots, Z_m$. We constructed a Directed Acyclic Graph (DAG), a graphical representation of the causal relationships among the variables, where the direction of the edges is determined by the sign and magnitude of the estimated CATE. For feature pair $(X, Y)$, the edge direction is $X \to Y$.

$$\tau(X) = \frac{1}{N} \sum_{i=1}^{N} \mathbb{E}\left[Y_i(1) - Y_i(0) \mid X_i\right]$$

For each confounder-treatment pair $(Z_j, X)$, a treatment model was fitted to predict the treatment variable $X$ using covariate features $Z_1, Z_2, \ldots, Z_m$. The treatment model is represented as:

$$\hat{X} = F_X(Z_1, Z_2, \ldots, Z_m)$$

where $F_X$ is the treatment model selected from the set of machine learning models. After fitting the treatment model, the outcome model was trained to predict the outcome variable $Y$ using the treatment variable $\hat{X}$ and covariate features $Z_1, Z_2, \ldots, Z_m$. The outcome model is denoted as:

$$\hat{Y} = F_Y(\hat{X}, Z_1, Z_2, \ldots, Z_m)$$

where $F_Y$ is the outcome model selected from a set of models. The residuals for both the treatment and outcome models were calculated as the difference between observed and predicted values

$$\mathrm{X}_{re} = X - \hat{X}$$

$$\mathrm{Y}_{re} = Y - \hat{Y}$$

After calculating the residuals, the residual model was trained to predict the outcome residuals using the treatment residuals

$$\mathrm{Y}_{re} = F_{\text{re}}(X_{\text{re}}) \tag{2}$$

where $F_{\text{re}}$ is chosen from a set of residual models.

$$R^2 = 1 - \frac{\sum_{i=1}^{n}(Y_i - \hat{Y}_i)^2}{\sum_{i=1}^{n}(Y_i - \bar{Y})^2}$$

This process is repeated for all feature pairs $(X_k, Y_l)$ in the dataset, where $k$ and $l$ index features. For each pair, DAG and validation $R^2$ scores were recorded. The best causal structure for each feature pair was identified based on a comparison of $R^2$ scores. An adjacency matrix $A$ is then derived from the DAG, where $A_{kl} = 1$ indicates a directed edge from $X_k$ to $Y_l$, and $A_{kl} = 0$ indicates no edge:

$$A_{kl} = \begin{cases} 1, & \text{if } X_k \to Y_l \\ 0, & \text{otherwise} \end{cases}$$

Adjacency matrix $A$ encapsulates the causal relationships among all the features, providing a comprehensive framework for causal inference. Bidirectionality was addressed by comparing the $R^2$ scores of the forward and reverse models. The business approach (3) uses pairs from domain knowledge whereas the union approach (4) uses the additional identified pairs. We created a DAG for both methods individually.

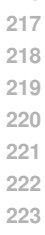
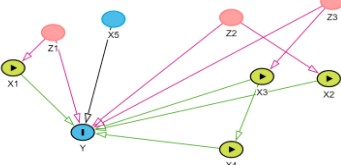
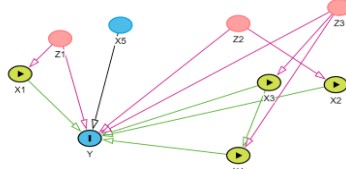

Figure 3: Directed Acyclic Graph (DAG) illustrating causal relationships between variables for the Business approach.

Figure 4: Directed Acyclic Graph (DAG) illustrating causal relationships between variables for the Union approach.

## 4.2 CAUSAL MODEL SELECTION USING SENSITIVITY ANALYSIS

In this section, we evaluate the effectiveness of the union approach by conducting experiments using real-world datasets. Our goal is to identify the most appropriate model combinations for treatment, outcome, and residual models with the observed confounders $Z$, treatment $X$, and outcome variable $Y$ within a Directed Acyclic Graph (DAG). Sensitivity analysis plays a crucial role by testing the stability of the models under random unobserved effects, thus providing a deeper understanding of causal impact assessment. A stable $R^2$ score is critical when introducing random confounders in the test data or shuffling the treatment, which nullifies the effect of causal relationships.

$$Y = \sum_{i=1}^{m} F_i^o(Z_i)$$

$$Y \subseteq (Z \cup X) \Rightarrow E(Y|Z, X)$$

where $F_i^o$ represents the individual spline functions for the confounder $Z_i$ that constitute the outcome model. Where $F_i^t$ is the representation of the individual spline functions for confounders $Z_i$ which constitute the treatment model.

$$X_t = \sum_{i=1}^{m} F_i^t(Z_i) \tag{4}$$

$$X \subseteq Z \Rightarrow E(X|Z)$$

The residual component captures what is left out by the outcome model. where $F_{\text{re}}$ is the representation of residual $X_{\text{re}}$ which constitutes our residual model.

$$Y_{\text{re}} = F_{\text{re}}(X_1^{\text{re}}, X_2^{\text{re}}, X_3^{\text{re}}, \ldots, X_m^{\text{re}})$$

Adjusted Outcome is the sum of outcome model prediction and residual model prediction. Where $F^o$ is the representation of the outcome model from the covariates and $F_{\text{re}}$ which constitutes our residual model.

$$Y^i = Y + Y_{\text{re}} = F^o(Z) + F_{\text{re}}(X_{\text{re}})$$

$$R_o^2 = 1 - \frac{E\left[(Y - E[Y|Z, X])^2 \big| Z, X\right]}{E\left[(Y - E[Y])^2\right]}$$

During sensitivity analyses, we randomly shuffled confounders and treatments individually within an unobserved dataset and observed their effect on the outcome variable. Shuffled confounders Goodness-of-Fit (R-squared) is calculated by using random methods of shuffling of confounders and observing their effect on the outcome variable as illustrated below.

$$Z_{\text{shuffled, test}} = \{Z_{\pi(1)}, Z_{\pi(2)}, \ldots, Z_{\pi(m)}\}$$

where $\pi$ denotes random permutations of confounder Z in testing sample of data

$$R_{rc}^2 = 1 - \frac{E\left[(Y - E[Y|Z_{\text{shuffled}}, X])^2 \big| Z_{\text{shuffled}}, X\right]}{E\left[(Y - E[Y])^2\right]}$$

Shuffled treatments R-squared is obtained by using random methods of shuffling of treatment and observing their effect on the outcome variable as illustrated below $X_{\text{shuffled, test}} = \{X_{\pi'(1)}, X_{\pi'(2)}, \ldots, X_{\pi'(m)}\}$

where $\pi'$ denotes random permutations of treatment $X$ in testing sample of data.

$$R_{rt}^2 = 1 - \frac{E\left[(Y - E[Y|Z, X_{\text{shuffled}}])^2 \big| Z, X_{\text{test}}\right]}{E\left[(Y - E[Y])^2\right]} \tag{5}$$

**Model Selection Using Neyman's Orthogonality Criterion** is applied to select the best model using a defined composite model score. Slight perturbations in conditional expectations $E[X \mid Z]$ do not significantly impact the composite model score. The composite model score integrates $R^2$ values to ensure Neyman's orthogonality:

$$\text{Model Score} = w_o \cdot R_o^2 + w_{rc} \cdot R_{rc}^2 + w_{rt} \cdot R_{rt}^2 \tag{6}$$

subject to the weights constraint and ensuring Neyman's orthogonality:

$$w_o + w_{rc} + w_{rt} = 1$$

$$E\left[\frac{\partial \text{Model Score}(E[X|Z] + \delta)}{\partial \delta} \bigg| E[X|Z]\right] = 0 \tag{7}$$

By incorporating the elements $Z$, $X$, and $Y$ in a Directed Acyclic Graph (DAG) context , we created a comprehensive causal framework that ensures robustness and correctness in causal inference and model selection. DAG helps visualize and clarify the dependencies, while set theory formalizes the definitions and relationships, and Neyman's orthogonality ensures the robustness of our estimations.

### 4.3 DETERMINING VARIABLE ATTRIBUTION WITH PARTIAL DEPENDENCY METHOD AND BASELINE SCALED MODELLING

**Baseline prediction and Secondary variables attribution**
**Input:** $(Z, X, O)$
**Output:** $B$, Secondary variables attributions
**Step 1:** Minimum influence
            Set features to minimal/median values
          $B_0 \leftarrow \text{predict}(0, \text{median})$
**Step 2:** Actual scenario
            Use actual input values
          $B_1 \leftarrow \text{predict}(\text{actual}, \text{actual})$
**Step 3:** Scaling factor $\text{Sc1} \leftarrow \frac{B_0}{B_0 + B_1}$
**Step 4:** Predicted baseline $B2 \leftarrow Y_{or} \times \text{Sc}$
**Step 5:** Recalculate residual $Y \leftarrow Y_{or} - B2$
**Step 6:** Non-Baseline attribution from causal model
          $NBA \leftarrow \text{Yo} + \text{Yr}$
**Step 7:** Sum of secondary variables attributions
          $TA \leftarrow 0$
          For each secondary variable Z,X :
          $A \leftarrow \text{calc\_att}(Z,X, \text{input}, \text{median})$
          $TA \leftarrow TA + A$
**Step 8:** Attribution for each secondary variable
          For each secondary variable Z,X :
          $A \leftarrow \text{calc\_att}(Z,X, \text{input}, \text{median})$
          $r \leftarrow \frac{A}{TA}$
          $AR \leftarrow r \times NBA$
**Step 9:** End

The baseline variable is denoted by $B$, with $B_0$ referring to the un-scaled baseline value, $B_1$ representing the actual scenario value, and $B_2$ indicating the scaled baseline value. The non-baseline attribution, which captures the influence of factors beyond the baseline, is represented by NBA. Additionally, the total attribution from secondary variables is captured by TA. To calculate the attribution from secondary variables, the function calc\_att is employed, while the function predict is

used to compute the baseline. The ratio of secondary variable attribution to the total is represented by $r$, with $A$ signifying the un-scaled secondary variable attribution, and AR corresponding to the scaled secondary variable attribution.

We implemented an output decomposition model comprising a robust baseline that removes seasonality, trends, and other effects, and a secondary variables causal model to understand the influence of primary variables (those directly impacting the outcome) and secondary variables (those with a causal relationship) . Given the confounders Z, treatment X and outcome variables Y and other variables, $O$.

### 4.3.1 BASELINE MODEL CONSTRUCTION

For baseline prediction, we start by defining the minimal influence scenario, $B_0$, where all primary variables $Z$, $X$, and $O$ have negligible or no impact. This is represented as $B_0 = f(\emptyset \cup \emptyset \cup \emptyset)$, implying that without the influence of these variables, the outcome would be minimal or zero.

The baseline model for the primary variables $Z$, $X$, and $O$, and the outcome $Y$ is a function of these sets:

$$Y = f(Z \cup X \cup O) \tag{8}$$

Baseline Prediction defines the minimal influence scenario $B_0$ where $Z$, $X$, and $O$ have minimal or zero influence:

$$B_0 = f(\emptyset \cup \emptyset \cup \emptyset) \tag{9}$$

In contrast, actual scenario, $B_1$, takes into account the presence and influence of all variables $Z$, $X$, and $O$:

$$B_1 = f(Z \cup X \cup O) \tag{10}$$

To normalize our predictions, we calculate the scaling factor $Sc1$, which adjusts the baseline prediction based on the expected values in both minimal and actual scenarios. Scaling factor $Sc1$ based on the expected values.

$$Sc1 = \frac{E(Y \mid Z = \emptyset, X = \emptyset, O = \emptyset)}{E(Y \mid Z = \emptyset, X = \emptyset, O = \emptyset) + E(Y \mid Z, X, O)}$$

Using this scaling factor, we calculate the predicted baseline $B_2$:

$$B_2 = E(Y \mid Z \cup X \cup O) \times Sc1 \tag{11}$$

This adjusted prediction reflects the expected outcome while accounting for the influence of the primary variables. Finally, we computed the residual by subtracting $B_2$ from the expected outcome to determine what remains unexplained by the baseline model

$$E(Y_o \mid Z \cup X \cup O) = E(Y \mid Z \cup X \cup O) - B_2 \tag{12}$$

### 4.3.2 SECONDARY VARIABLES ATTRIBUTION

Non-Baseline Attribution $E(\text{NBA} \mid Z, X)$ for secondary variables $Z$ and $X$ is computed by summing up the contributions from the outcome model $E(Y_{\text{out}} \mid Z, X)$ and the residual model $E(Y_{\text{res}} \mid Z, X)$. This provided a measure of the influence of these secondary variables on the outcome.

$$E(\text{NBA} \mid Z, X) = \sum_{Z, X \in Z \cup X} (E(Y_{\text{out}} \mid Z, X) + E(Y_{\text{res}} \mid Z, X)) \tag{13}$$

For each secondary variable $Z_i$ or $X_i$, the attribution $A_{Z_i}$ or $A_{X_i}$ is calculated by setting $Z_i$ or $X_i$ to its actual value while keeping the others minimal:

$$E(A_{Z_i} \mid Z_i) = E(Y \mid Z_i, X = \emptyset, O = \emptyset) \tag{14}$$

$$E(A_{X_i} \mid X_i) = E(Y \mid X_i, Z = \emptyset, O = \emptyset) \tag{15}$$

Total attribution $E(\text{TA} \mid Z, X)$ from all secondary variables $Z$ and $X$:

$$E(\text{TA} \mid Z, X) = \sum_{Z, X \in Z \cup X} (E(A_{Z_i} \mid Z_i) + E(A_{X_i} \mid X_i))$$

The attribution for each secondary variable $Z_i$ or $X_i$ is further expressed as a ratio to provide a proportional contribution to the non-baseline attribution:

$$E(\text{AR}_{Zi} \mid Z) = \frac{E(A_{Z_i} \mid Z_i)}{E(\text{TA} \mid Z, X)} \times E(\text{NBA} \mid Z) \tag{16}$$

$$E(\text{AR}_{Xi} \mid X) = \frac{E(A_{X_i} \mid X_i)}{E(\text{TA} \mid Z, X)} \times E(\text{NBA} \mid X) \tag{17}$$

The expected value of the outcome given the primary variables $Z$, $X$, and $O$ is denoted as $E(Y \mid Z, X, O)$. We also consider the expected attribution due to specific secondary variables $Z_i$ or $X_i$, conditioned on minimal influence from other primary variables, represented as $E(AZ_i \mid Z_i)$ and $E(AX_i \mid X_i)$, respectively. Further, the expected non-baseline attribution given the secondary variables $Z$ and $X$ is denoted as $E(NBA \mid Z, X)$, while the expected sum of all secondary variables' attributions, given $Z$ and $X$, is expressed as $E(TA \mid Z, X)$. In addition, we calculate the expected scaled attribution for secondary variables $X_i$ and $Z_i$, represented as $E(ARX_i \mid X)$ and $E(ARZ_i \mid Z)$, respectively.

## 5 EXPERIMENTS

In this section, we evaluate the effectiveness of our approach with attributions for the variables and their corresponding values from the baseline single-stage best model.

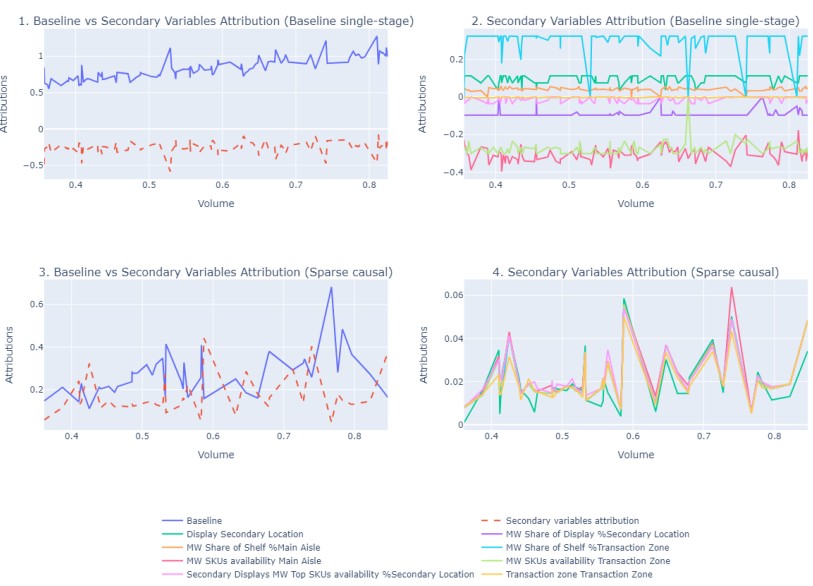

Figure 5: Baseline and secondary variables attributions for baseline single-stage model and sparse causal model

### 5.0.1 COMPARISON WITH BASELINES

**Dataset:** We utilized the Perfect Store Key Performance Indicator (KPI) survey data, along with sales data for fast-moving consumer goods (FMCG) comprising 43,000 stock-keeping units (SKUs). The dataset was randomly split, with 80% allocated for training and the remaining 20% reserved for testing. The model's performance was assessed using the $R^2$ score on the test dataset.

The attributions derived from the sparse causal model demonstrate significant reliability, addressing the issue of negative secondary variable attribution. This model also yields a higher baseline compared to the traditional single-stage model Enouen & Liu (2022); Imbens (2004); Bergstra et al.

Table 1: Performance comparison with other models

| Dataset | Model | Category | Channel | R-squared (Train) | R-squared (Test) |
|---------|-------|----------|---------|-------------------|------------------|
| | Linear Regression | Express A | Express A | 0.0527 | 0.0476 |
| | Ridge | Chocolate | Express A | 0.0505 | 0.0384 |
| Perfect | Lasso | Chocolate | Express A | 0.0000 | -0.0003 |
| Store | Random Forest | Chocolate | Express A | 0.8346 | -0.0054 |
| (KPI) | Gradient Boosting | Chocolate | Express A | 0.6636 | -0.1058 |
| Survey | XGBoost | Chocolate | Express A | 0.9999 | -0.1669 |
| Data | KNN Regression | Chocolate | Express A | 0.2615 | -0.0713 |
| | **Sparse Causal Model** | Chocolate | Express A | **0.5797** | **0.5771** |

Table 2: Best model selection from sparse causal framework

| Category | Channel | Model Combinations (3-stages) | $R^2$ (Train) | $R^2$ (Test) |
|----------|---------|-------------------------------|---------------|--------------|
| **Chocolate** | **Express A** | **Linear Regression, Linear Regression, XGBoost** | **0.5797** | **0.5771** |
| Chocolate | Express A | Linear GAM, XGBoost, Linear GAM | 0.4961 | 0.0734 |
| Chocolate | Express A | Linear GAM, Linear GAM, Linear GAM | 0.6110 | 0.1683 |
| Chocolate | Express A | Linear GAM, Linear Regression, Linear GAM | 0.3195 | 0.3184 |

(2011). Multiple experiments across various granularities further support that our model outperforms others in discovering causality, optimizing model combinations through exploitation methods, and utilizing sensitivity scores to accurately infer causality and ensure the smoothness of attributions. Additionally, traditional models often suffer from model fit, particularly when dealing with sparse observations that exhibit causal relationships.

## 6 CONCLUSION

In conclusion, this research presents a novel framework for causal discovery and attribution that effectively addresses the challenges posed by sparse real-time datasets and limited feature knowledge. Our integrated "union" approach combines domain knowledge with the discovery of new causal relationships, resulting in more accurate and robust Directed Acyclic Graph (DAG) constructions. By blending parametric and non-parametric models with advanced sensitivity analyses, our framework surpasses traditional methods, ensuring correct causal identification and controlling the family-wise error rate. It enhances inference reliability by isolating robust causal models, providing a flexible tool for researchers across various fields. Our approach improves causal inference accuracy by 53%, particularly in addressing attribution challenges in sparse datasets.

Future research could further refine and expand upon this framework, particularly by exploring its applicability to other types of data and integrating additional techniques, such as bootstrapping, to enhance causal discovery across diverse sparse datasets. While our experiments on real-world datasets validate the effectiveness of our approach, additional empirical studies across different domains would provide deeper insights into its broader applicability. This study represents a significant advancement in the field of causal discovery and attribution, offering a practical and theoretically sound solution for improving the accuracy and robustness of causal inferences in complex data environments.

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
