# OpenReview forum: "Sparse Causal Model: A Novel Approach for Causal Discovery and Attributions on Sparse Dataset"
_ICLR.cc/2025/Conference — Submitted to ICLR 2025_

### Official Review · Reviewer_M5oA · 2024-10-31

**Soundness:** 1
**Presentation:** 2
**Contribution:** 1
**Rating:** 3
**Confidence:** 4

**Summary:**

This paper introduces an approach to tackle the challenges of causal modeling and attribution on sparse and non-continuous data with limited feature knowledge. The new approach combines multiple machine learning models to select the optimal fitting model for casual discovery and attributions. The R-square scores obtained from the experimental results show that the method presented in the paper is 53% higher compared to the single-model fitting result.

**Strengths:**

1. The paper presents a novel causal modeling and attribution framework specifically designed to address the challenges of sparse and non-continuous data.
2. The paper employs partial dependence methods to gain deeper insights into variable attribution, significantly improving the model's goodness of fit.

**Weaknesses:**

**Main argument**



The paper does not provide a solid theoretical basis for the proposed method. The combination of different models and the selection of confounder-treatment pairs seem rather ad-hoc without a clear theoretical justification. There is a lack of in-depth discussion on why these specific models and techniques are chosen and how they interact to achieve the claimed results.



The experiments are not comprehensive enough to support the superiority of the proposed method. The comparison with baselines is limited to two specific datasets (Perfect Store Key Performance Indicator (KPI) survey data and sales data for fast-moving consumer goods (FMCG)), and it is not clear if the results can be generalized to other datasets. The performance evaluation using only the score may not be sufficient to fully assess the quality of the causal model. There is also a lack of comparison with state-of-the-art methods in the field of causal discovery and attribution for sparse and non-continuous data.

There are some theoretical errors and many missing details:

1. No detailed proof that correlation path coefficients can identify confounder-treatment pairs, and in other words, why did the situation of treatment - treatment pairs with top-k high correlation coefficients not occur?
2. How did you distinguish whether a negative CATE represents a negative effect or a reverse direction of causality?
3. How can it be proven that a high R2 score resulting from a high goodness-of-fit of the correlation model indicates that the recovered causal relationship is correct?
4. The paper lacks identifiability proof.
5. Why only R-squared ？
6. Why is a 3-stage model chosen instead of 4-stage, 5-stage, or other multi-stage models?
7. Why are the Linear Regression, Linear Regression and XGBoost models selected rather than the Random Forest, SVM or other model combinations?
8. Why only Chocolate among Categories?
9. How did you demonstrate that the approach is adaptable to dynamic changes?
10. The analysis of the experimental results in this paper is badly incomplete.
11. The paper lacks hyperparameter sensitivity analysis.

There are some inaccuracies in this paper. For example, the process of identifying confounder-treatment pairs and constructing the Directed Acyclic Graph (DAG) could have been explained in more detail to enhance the clarity of the method. Additionally, the introduction to the dataset is rather lacking. Key details such as the source of the data, the size and characteristics of the dataset have not been adequately described. Due to the lack of information about the dataset, such as whether it is non-continuous data, whether it is sparse, whether it is dynamic, etc., it is difficult for readers to perceive the importance of this research.

**The paper has many imprecise parts, here are a few:**

1. It is not clear that Y is the outcome variable, but in section 4.2, why Y ⊆ (Z ∪ X).
2. Not sure what you mean with X ⊆ Z in section 4.2 for the reason that Z is the confounder variables set and X is the treatment variables set.
3. It is confused whether F_x, F_Y and F_re are selected from the same set of models.
4. It is not clear what Figures 3 and 4 are intended to illustrate.
5. It is not clear what the legends such as MW and so on in Figure 5 are intended to indicate.
6. It should be explained in detail about figures and tables, the data content presented, and how they support the research conclusions of the paper.

**Things to improve the paper that did not impact the score:**

1. Express A appears in the Category column in Table 1.
2. Although concepts like "Neyman's Orthogonality Criterion" and "Directed Acyclic Graph (DAG)" may be common in related fields, more detailed explanations could be considered to facilitate better understanding for readers from different backgrounds.
3. When introducing the method of selecting confounder-treatment pairs based on correlation path coefficients, some practical examples could be added to illustrate how to calculate and select these pairs.
4. The titles and axis labels of the charts should also be more clear and accurate, so that readers can quickly understand the core information of the charts.

**Questions:**

See the weaknesses above.

**Details Of Ethics Concerns:**

NA.

---

### Official Review · Reviewer_ebaM · 2024-11-01

**Soundness:** 2
**Presentation:** 2
**Contribution:** 3
**Rating:** 5
**Confidence:** 3

**Summary:**

This paper introduces a novel causal discovery framework called the "Sparse Causal Model" designed for analyzing sparse and non-continuous datasets with limited feature knowledge. The framework aims to overcome the limitations of traditional causal inference methods when dealing with sparse data by integrating domain knowledge with newly discovered causal relationships in a DAG-based approach.

**Strengths:**

1. The authors have conducted comprehensive experimental validation using real-world data, demonstrating practical applicability.
2. The authors have structured their paper in a logical and easy-to-follow manner, with clear problem formulation and methodology sections.

**Weaknesses:**

1. The authors have not provided sufficient theoretical guarantees for their proposed framework's convergence and stability.
2. The authors have not included comprehensive comparisons with state-of-the-art causal discovery methods, making it difficult to assess relative performance.
3. The paper lacks an analysis of computational complexity and scalability considerations.
4. The authors have provided insufficient information about their hyperparameter selection process for the different models.

**Questions:**

1. Can the authors provide theoretical guarantees for the convergence of their three-stage model?
2. How does the computational complexity scale with the number of variables and observations?
3. What is the justification for the specific weights chosen in the Model Score calculation (Equation 6)?
4. How does the framework perform on other types of sparse datasets besides the Perfect Store KPI data?
5. What is the sensitivity of the results to different choices of k in selecting the top confounder-treatment pairs?
6. Could the authors provide ablation studies to demonstrate the contribution of each component in the three-stage model?

---

### Official Review · Reviewer_7MhK · 2024-11-02

**Soundness:** 1
**Presentation:** 1
**Contribution:** 1
**Rating:** 1
**Confidence:** 3

**Summary:**

This paper introduces a novel causal discovery framework designed to tackle the challenges of causal modeling and attribution in sparse, non-continuous datasets with limited feature knowledge. The authors propose a three-stage approach that combines multiple regressors (likelihood-based, tree-based, and Generalized Additive Models) and leverages a Directed Acyclic Graph (DAG) to discover causal relationships by identifying confounder-treatment pairs. The framework includes a unique Model Score that incorporates sensitivity analysis through random shuffling of confounders and treatments to select the optimal model, along with a partial dependency approach to understand variable attribution. The methodology also implements a baseline scaled modeling technique that separates direct and indirect effects while removing seasonality and trends. When tested on Perfect Store Key Performance Indicator survey data, their approach demonstrated a 53% improvement in R² score compared to traditional methods, particularly excelling in addressing attribution challenges in sparse datasets.

**Strengths:**

I struggle to identify any.

**Weaknesses:**

This paper lacks clarity on all fronts. There are too many examples to list them all. Here are a few:
- L33: "traditional methods struggle with sparse datasets" This is not supported by evidence. Also, what is meant by sparse?
- Sec. 3 on Problem Formulation: Almost all assumptions are missing.
- Sec 4.0.1: The method combines linear regression, XG boost and GAMS. Why? They are all regression models, why would they need to be combined.

I'm afraid that in order to accept the paper, much more than minor corrections are necessary. Therefore, I suggest rejecting it.

**Questions:**

None

---

### Official Review · Reviewer_RRfi · 2024-11-04

**Soundness:** 3
**Presentation:** 1
**Contribution:** 1
**Rating:** 3
**Confidence:** 3

**Summary:**

This paper presents a framework for causal discovery in sparse datasets, addressing challenges in dynamic causal relationship modeling and attribution in the presence of limited feature knowledge. The approach introduces a three-stage causal model that integrates likelihood-based, tree-based, and GAM. Additionally, sensitivity analysis for random shuffling of confounders is presented. Experimental results show that the proposed framework improves $R^2$ scores over traditional methods.

**Strengths:**

- The comparison with various baseline models highlights the effectiveness of the sparse causal model, and the improvement in R² scores is quantitatively significant.

- The paper present a sensitivity analysis strengthens model selection ensuring the model is reliable under diverse data configurations in real-world scenario.

**Weaknesses:**

- Although the model is showing to be useful, the multi-stage framework involving various regressors may reduce transparency in attribution results. A intuitive and deeper interpretation of different components could help reader to better understand.

- The paper could benefit from an analysis of the computational complexity of the proposed approach. Integrating multiple models within a causal discovery framework may be computationally expensive.

- One of the main focus of this paper is about the sparse dataset, yet the paper does not provide a rigorous definition or characteristics of sparsity in this context. Sparse data could mean different things depending on the application. The paper could be benefit from a clear definition for the sparsity and as well as a motivation discussion for how existing method fails under such setting.

- Similarly, the paper mentions the model could adapts to dynamic changes in causal relationships, but does not explain what this entails.

- Given the vague definitions, the problem itself lacks strong motivation. For example, without a clear understanding of "sparse dataset" or "dynamic changes" the readers cannot fully appreciate why these challenges are significant or why the proposed model is suitable for addressing them. The paper should emphasize on these aspects where traditional methods fail, showing where this model could provide significant improvements.

- The paper does not seem to provide a strong theoretical or practical contribution in the research area. The paper could be improved by strengthen the theoretical or empirical contributions, such as through novel model design or solid theoretical foundation of the problem.

**Questions:**

- What is the main motivation for the proposed method?

- The variable attribution should be defined in Section 4.3 before it appears in the algorithm.

- Maybe include references for the causal attributions and causal mediation analysis.

---

### Meta-Review · Area_Chair_ffmM · 2024-12-21

**Metareview:**

The authors introduce the "sparse causal model", a causal discovery framework for causal modeling and attribution in sparse datasets with limited feature knowledge. Reviewers are unanimously in favor of rejection, and the authors did not provide a response.

**Additional Comments On Reviewer Discussion:**

No author response.

---

### Decision · Program_Chairs · 2025-01-22

Reject